# Sulfur-Containing Metabolites from Marine and Terrestrial Fungal Sources: Origin, Structures, and Bioactivities

**DOI:** 10.3390/md20120765

**Published:** 2022-12-07

**Authors:** Zhaoming Liu, Mingqiong Li, Shuo Wang, Huibin Huang, Weimin Zhang

**Affiliations:** State Key Laboratory of Applied Microbiology Southern China, Guangdong Provincial Key Laboratory of Microbial Culture Collection and Application, Institute of Microbiology, Guangdong Academy of Sciences, 100 Central Xianlie Road, Yuexiu District, Guangzhou 510070, China

**Keywords:** sulfur-containing natural products, fungi, chemical structures, bioactivities

## Abstract

Organosulfur natural products (NPs) refer to the different kinds of small molecular-containing sulfur (S) elements. Sulfur-containing NPs tightly link to the biochemical processes and play an important role in the pharmaceutical industry. The majority of S-containing NPs are generally isolated from Alliaceae plants or bacteria, and those from fungi are still relatively rare. In recent years, an increasing number of S-containing metabolites have been discovered in marine and terrestrial fungi, but there is no comprehensive and targeted review to summarize the studies. In order to make it more straightforward to better grasp the fungal-derived S-containing NPs and understand the particularity of marine S-containing NPs compared to those from terrestrial fungi, we summarized the chemical structures and biological activities of 89 new fungal-derived S-containing metabolites from 1929 when the penicillin was discovered to the present in this current review. The structural and bioactive diversity of these S-containing metabolites were concluded in detail, and the preliminary mechanism for C-S bond formation in fungi was also discussed briefly.

## 1. Introduction

Sulfur (S) is one of the most important elements for life, and numerous biochemical processes are tightly linked to this element; for example, cysteine is essential in protein synthesis and in protein-folding pathways [1]. Organosulfur natural products (NPs) refer to the different kinds of natural products containing sulfur elements, such as thiols, thioesters, sulfoxides, etc. [2,3,4,5], which play an important role in the pharmaceutical industry. It was reported that there are 41 sulfur-containing drugs that were driven or modified from organosulfur NPs appearing in the Top 200 Pharmaceuticals by Retail Sales in 2019 worldwide [6]. As is known, penicillin, cephalosporine, and trabectedin (ET-743) are S-containing NPs widely used as clinical drugs, while some chemically synthetic drugs inspired by NPs, such as phthalascidin and quinupristin, are also used to treat different kinds of diseases. In recent years, various S-containing metabolites have been isolated from plants, animals, or microorganisms with diverse biological activities, such as anti-inflammatory [7], anticancer [8], and plant defense [9]. The majority of S-containing NPs are generally isolated from *Alliaceae* plants or bacteria, and those from fungi are still relatively rare. Several reviews have been published to summarize the S-containing NPs from *Allium* spp. as well as their bioactivities [10,11]. However, the only review referring to fungal-derived S-containing NPs is reported by Shao CL and his co-workers. They concluded that 484 S-containing metabolites from marine microorganisms covered 44 fungal metabolites (excluding thiodioxopiperazines) from January 1987 to December 2020 [12]. However, there is no comprehensive review specially focusing on fungal organosulfur metabolites from both marine and terrestrial environments to date.

In order to make it more straightforward to better grasp the fungal-derived S-containing NPs and understand the specialty of marine S-containing natural products compared to those from terrestrial fungi, we discussed the chemical structures and bioactive properties of the new non-sulfated S-containing NPs discovered from both marine and terrestrial fungi from 1929 when the penicillin discovered to the present in the current review. Thiodioxopiperazines (TDPs) are the most abundant S-containing metabolites from nature, the sulfur of which was proven to be driven by glutathione (GSH). Two systematic reviews carried out by Jia’s and Li’s groups provided the summary of 166 naturally occurring diketopiperazine alkaloids from 1944 to 2015 and 83 irregularly bridged epipolythiodioxopiperazines from nature, respectively [13,14]. The other review published by Shao et al. concluded that the sulfur-containing NPs from marine microorganisms from 1987 to 2020 included 174 TDPs [12]. Since the TDPs are too much abundant in fungi, and the above reviews have given a comprehensive summary of this class of metabolites, we do not provide a detailed discussion in this review. The polypeptides that were constructed by S-containing amino acids and sulfated metabolites that did not construct C-S bonds were also excluded from this review. As a result, a total of 89 new S-containing metabolites (more than half of which were from marine resources), which can be divided into five main groups according to their structure features, have been summarized in this review (Figure 1). The major class is polyketides, contributing 41% of the total number of metabolites. Though the macrolides and cytochalasins belong to the polyketides, they are still summarized as a separate class in this review, which accounts for 27% and 10% of compounds, respectively. The alkaloids that account for 21% are the most abundant class except for polyketides. Moreover, only one S-containing terpenoid (phomenone A) was isolated from a mangrove-derived fungus, *Penicillium* sp., to date, which exhibited a weak antibacterial effect against *Escherichia coli* [15]. Herein, we present the isolation, structural diversity, and bioactivities of these S-containing metabolites in detail. A brief summary and outlook of the biosynthetic mechanism of how the fungi introduce the S atom into their metabolites are discussed at the end of the review.

## 2. Isolation, Structural Features, and Bioactivities of S-Containing NPs

### 2.1. Polyketides

Polyketides are the largest group among S-containing NPs from fungi. Though the macrolides and cytochalasins are excluded from the class of polyketides, there are 36 polyketides, including chromones, xanthones, quinones, benzoic acid, and isocoumarin, isolated from marine and terrestrial fungi, which were classified into thioether, thioester, sulfoxide, and sulphonyl according to the types of sulfur introduction (Figure 2, Figure 3, Figure 4, Figure 5 and Figure 6). 

#### 2.1.1. Thioether-Containing Polyketides

*Chaetomium* is a potential fungal genus that can produce diverse bioactive metabolites. The chemical investigation of the solid fermented culture of the strain *Chaetomium seminudum* purchased from Shaanxi Institute of Microbiology led to the isolation of two cysteine-derived chromones, chaetosemins A and B (**1** and **2**) [16]. The epimerization at C-4′ was because both L- and D-cysteine participate in the biosynthetic pathway. Interestingly, only **2** exhibited antifungal activities against the phytopathogenic fungi *Magnaporthe oryzae* and *Gibberella saubinettii* with MIC values of 6.25 and 12.5 μM, respectively. Chromosulfine (**3**) is a novel cyclopentachromone sulfide isolated from the mutated marine-derived fungus *Penicillium purpurogenum,* which was obtained by treating the wild strain with 6.7 mg/mL neomycin in 67% DMSO (Figure 2). The absolute configuration in the core skeleton was established by the ^1^H-coupled ^13^C NMR spectral analysis and ECD calculation. Compound **3** exhibited moderate cytotoxicity against human cancer cell line HL-60 with the IC_50_ value of 16.7 μM [17]. Four chromone cyclothioether derivatives, coniothiepinols A (**4**) and B (**5**), coniothienol A (**6**), and preussochromone A (**7**), were isolated from the endophytic fungi *Coniochaeta* sp. and *Preussia africana* (Figure 2). All of them constructed a C_5_ unit at the C-3 position of the chromone core and cyclized to C-2 via a S atom in different ways. In the bioassays, coniothienol A (**6**) showed significant antibacterial activities against the Gram-positive bacteria *E. faecium* and *E. faecalis* with IC_50_ values of 2.00 and 4.89 µg/mL, while coniothiepinol A (**4**) exhibited moderate activities against not only the above bacteria (3.93 and 11.51 µg/mL, respectively) but also the plant-pathogenic fungus *F. oxysporum* (13.12 µg/mL). Preussochromone A (**7**) was cytotoxic to the human cancer cell lines A549, Hela, and HCT116 with IC_50_ values of 8.34, 25.52, and 25.87 µM, respectively [18,19].

**Figure 2 marinedrugs-20-00765-f002:**
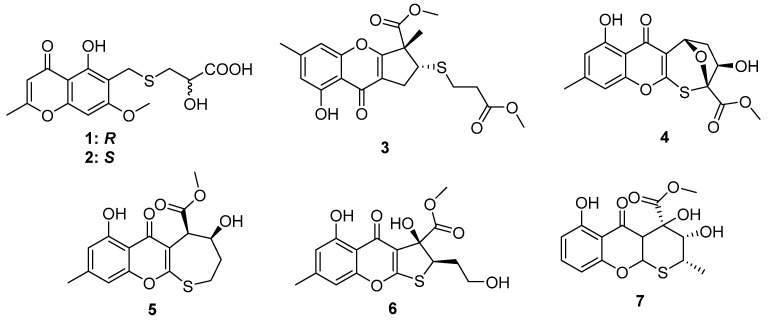
Chemical structures of **1**–**7**.

**Figure 3 marinedrugs-20-00765-f003:**
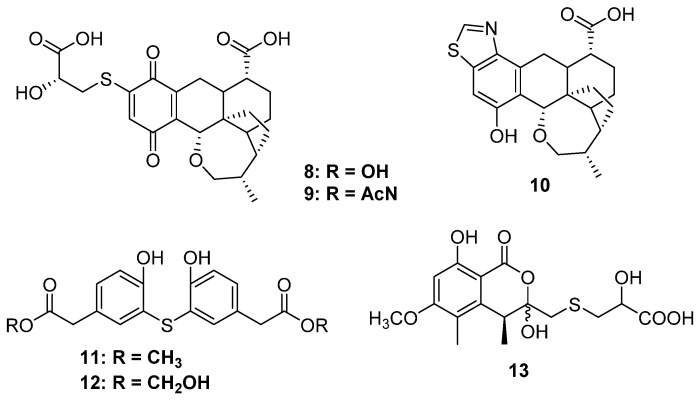
Chemical structures of **8–13**.

Thiopleurotinic acids A (**8**) and B (**9**) as well as pleurothiazole (**10**) were three quinones derivatives bearing a complicated bicyclic [4.2.1] moiety discovered from the fungus *Hohenbuehelia grisea*, which was collected from decaying wood (Figure 3). Feeding of the [U-^13^C_3_ ^15^N]-L-cysteine indicated that the 2-hydroxy-3-mercaptopropanoic acid moiety is derived from cysteine, and the absolute configuration at the side chain was deduced to be *S* unambiguously [20]. Only **92** and **10** possessed weak inhibitory activity against yeasts, such as *Candida tenuis*, *Pichia anomala*, and *Rhodotorula glutinis,* without any cytotoxicity. A strain of *Penicillium copticola* PSU-RSPG138 collected from soil produced two phenyl sulfide derivatives, penicillithiophenols A (**11**) and B (**12**), and another terrestrial fungus, *Aspergillus banksianus,* yielded an isocumarin thioether banksialactone E (**13**). All of them showed no inhibitory activities against the tested microorganisms or cells [21,22] (Figure 3).

**Figure 4 marinedrugs-20-00765-f004:**
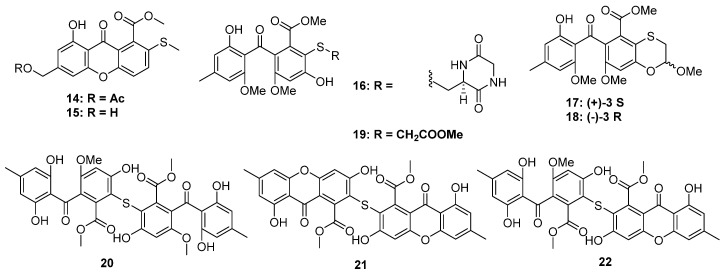
Chemical structures of **14**–**22**.

Xanthones are common in thioether-containing polyketides since the core skeleton possesses different degrees of substitution. Two xanthone-methyl sulfide derivatives, sydoxanthones A and B (**14** and **15**), were isolated from *Aspergillus sydowii,* which was collected from Chinese liverwort *S. ciliate* (Figure 4) [23], of which only **15** exhibited potential immunosuppressive activities against Con A or LPS-induced proliferation of mouse splenic lymphocytes. The chemical investigation of a cave soil-derived fungus, *Aspergillus fumigatus* GZWMJZ-152, led to the isolation of three hydrolyzed xanthones (**16**–**19**) (Figure 4). Compound **16** represents a special xanthone-diketopiperazine hybrid thioether. Compound **19** exhibited significant antioxidant capacity with an ORAC index of 1.65 μmol TE/μmol. Moreover, in the H_2_O_2_-induced oxidative injury of PC12 cells, the enantiomerically pure **17** and **18** exhibited the same protective effects as their racemic mixtures [24]. Three xanthone dimers belonging to the sulochrin family (**20** to **22**) were isolated from an *Alternaria* sp. collected from a Hawaiian soil sample (Figure 4). Perhaps the latter two compounds, dioschrin (**21**) and castochrin (**22**), are artifacts since **20** was susceptible to intramolecular cyclization under aqueous conditions. Compounds **20** to **22** showed significant anti-MRSA activity with MIC values of 2.9, 3.2, and 2.0 μg/mL, which were close to the positive control chloramphenicol (1.6 μg/mL) [25].

#### 2.1.2. Thioester-Containing Polyketides

The thioester moiety is unusual in S-containing NPs. To date, only a series of methyl sulfide benzoate derivatives were discovered from two marine fungi, including eurothiocins A and B (**23** and **24**), from soft coral-derived fungus *Eurothium rubrum,* as well as the eurothiocins C–H (**25** to **30**) from deep-sea-derived fungus *Talaromyces indigoticus* (Figure 5). Most of them were substituted by an isopentenyl unit at C-6 of the benzene ring. Because of the different oxidation and cyclization way of the isopentenyl, their structures could construct either an isopentenyl benzoate framework or a benzofuran core. In the bioassays, **23**, **24**, **26**, **28,** and **29** exhibited potential α-glucosidase inhibitory activities, and the theoretical docking study of these compounds to the α-glucosidase protein suggested that a hydrophilic terminal of the isopentenyl group was important to the bioactivities since the terminal hydroxyl group could form hydrogen bonds to the residues in the active docking pocket, such as Asp215, Val216, etc. [26,27].

**Figure 5 marinedrugs-20-00765-f005:**
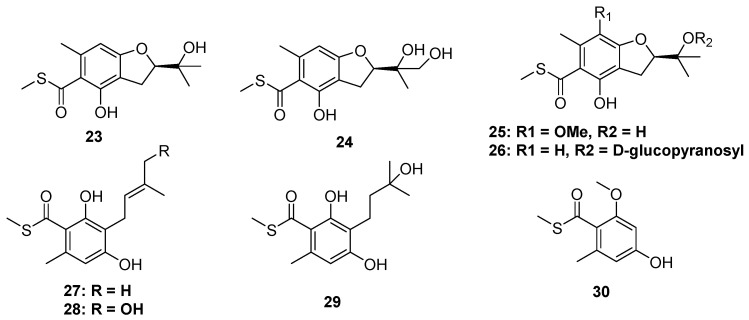
Chemical structures of **23–30**.

#### 2.1.3. Sulfinyl and Sulfonyl-Containing Polyketides

Sulfinyl (sulfoxide) or sulfonyl-containing metabolites are also very rare in fungi. Though a series of sulfinyl or sulfonyl-containing 6-methylthiochroman-4-one derivatives were obtained by biotransformation using *Trichoderma viride* [28], only seven metabolites were produced by five strains of fungi. The fungus *Aspergillus banksianus* yields not only isocumarin-cysteine thioether banksialactone E (**13**) but also its oxidative derivative, banksialactone F (**31**), containing isocumarin-methyl sulfoxide (Figure 6) [22]. Another coral-associated fungus, *Pseudallescheria boydii,* could produce the sulfinyl-containing metabolites (**32**) as well as a known analog (**33**), both of which constructed a thiopyran-S-oxide moiety [29]. The sulfoxide group generated an additional stereogenic center because of the existence of lone pair electrons. Even though several natural products constructing a similar unit were obtained from garlic plants [30,31,32,33,34,35,36], their absolute configuration was almost unidentified. In the structural identification of **32** and **33**, a theoretical ECD calculation was carried out to directly establish the absolute configuration of the sulfoxide group, which makes a major contribution to the ECD spectrum [29]. The strain *Aspergillus fumigatus* GZWMJZ-152 not only produced xanthone thioether but also yielded a racemic mixture of xanthone-methyl sulfinyl ((+)-**34** and (−)-**34**). The absolute configuration of sulfinyl was also established by the ECD calculations. Like the other isolated xanthone derivatives from this strain, the optical pure and enantiomer mixtures of **34** exhibited antioxidative activities [24]. (±)-Prunomarin A (**35**) was isolated from the endophytic fungus *Phomopsis prunorum*, which represented the first example of sulfoxide-containing isocoumarins featuring a 4,5-fused dihydrothiopyran 1-oxide moiety from endophytic fungi [37].

**Figure 6 marinedrugs-20-00765-f006:**
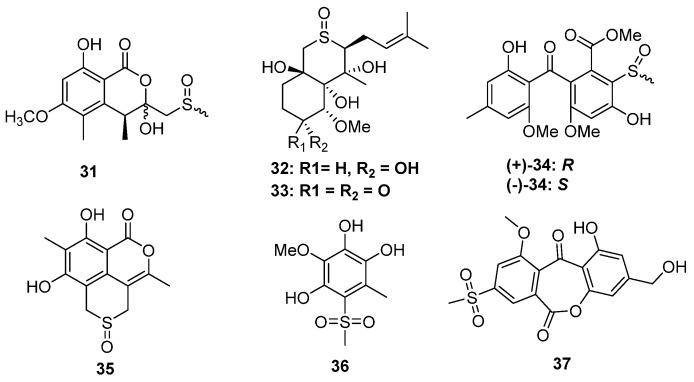
Chemical structures of **31**–**37**.

Based on the epigenetic modification strategy, the mangrove-derived fungus *Neosartorya udagawae* was cultivated with the DNA methyltransferase inhibitor 5-azacytidine. As a result, two new polyketides, 3-methoxy-6-methyl-5-(methylsulfonyl)-benzene-1,2,4-triol (**36**) and neosartoryone A (**37**), constructing a rare methylsulfonyl group, were isolated (Figure 6). Induced by 5-azacytidine, the strain *N. udagawae* could utilize the DMSO as a sulfur source to form the sulfonyl moiety, and this is the first report of a fungus that can achieve such a sulfonylation-like modification of natural products. In the bioassays, **37** could decrease the lipid accumulation elicited by oleic acid at a concentration of 10 μM without any toxicity [38].

### 2.2. Macrolides

Macrolides are actually a kind of polyketide driven from the C2 or C3 units. In this review, the macrolides containing the S atom are classified into a separate category since they all present a 12-membered ring lactone skeleton. Except for the dimers, nearly all the isolated macrolides bear or are driven from a cysteine moiety via the S atom at the C-2 or C-3 position, while only two analogs are thioglycolic acid thioether derivatives (**50** and **59**). As for the fungal resource, all the fungi producing S-containing macrolides were discovered in marine environments. The details are summarized below (Figure 7, Figure 8 and Figure 9):

#### 2.2.1. Dihydroxyphenylacetic Acid Macrolides

Dihydroxyphenylacetic acid lactones (DALs) are a class of familiar macrolides from nature, and fungi are notable producers of them. However, the S-containing DALs are unprecedented (Figure 7). De Castro and his co-workers collected *Penicillium* sp. DRF2 from the sponge *Dragmacidon reticulatum,* which could produce new DALs. [39]. By using the statistical experimental design methodology and chemometric analysis to improve the secondary metabolites production of DRF2, six new cysteine-combined DALs belonging to the curvularines family were isolated (**38**–**43**). Moreover, isotope-feeding experiments with [U-^13^C_3_^15^N]-L-cysteine confirmed the presence of 2-hydroxy-3-mercaptopropanoic acid residue and oxidized sulfoxide in these compounds [40]. Moreover, sumalarins A–C (**44**–**46**) are also three new cysteine additive DALs isolated from the mangrove-derived fungus *Penicillim sumatrense* MA-92. All of them exhibited significant cytotoxicity against seven tested human cancer cell lines [41].

**Figure 7 marinedrugs-20-00765-f007:**
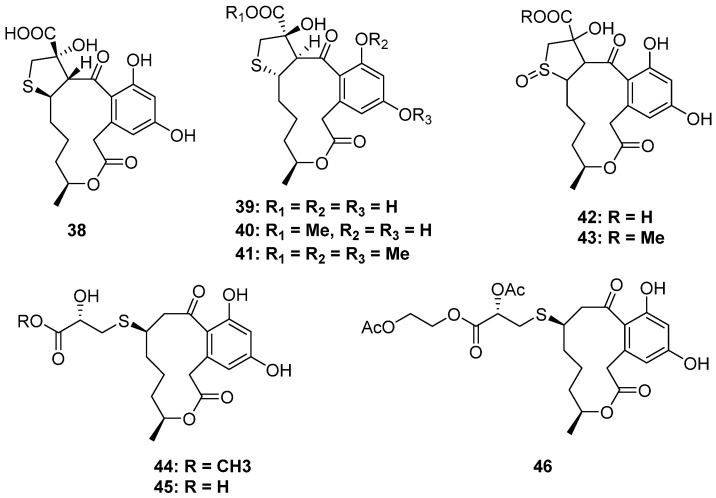
Chemical structures of **38–46**.

#### 2.2.2. 2-Thioether-Substituted Macrolides

Cladosporioidin A (**47**) is the most complicated S-containing macrolide discovered from a marine cold-seep-derived fungus, *Cladosporium cladosporioides,* to date, which constructed a tricyclic system containing a 12-membered lactone, tetrahydrothiophene and a peroxy lactone (Figure 8). The relative configuration at C-11 was established by ^13^C NMR calculations and DP4 simulations since it was far away from the bicyclic core and was hard to identify through the NOESY spectrum. Compound **47** exhibited weak antibacterial activity against three marine-derived bacteria [42].

**Figure 8 marinedrugs-20-00765-f008:**
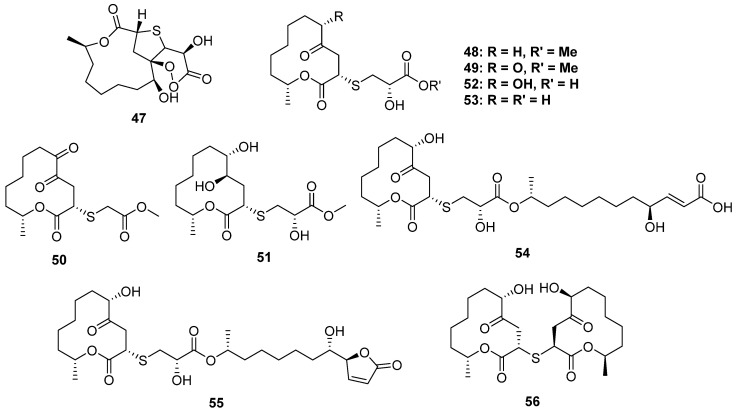
Chemical structures of **47**–**56**.

The fungal species of *Cladosporium* collected from mangrove plants are the largest producer of S-containing macrolides (Figure 8). The strain of *Cladosporium cladosporioides* MA-299 from *Bruguiera gymnorrhiza* yielded four cysteine derivative 12-membered ring lactones thiocladospolides A–D (**48**–**51**), and another mangrove-endophytic fungus, *Cladosporium oxysporum,* also yielded four analogs, thiocladospolides G–J (**52**–**55**), as well as a 2-thioether-dimer, thiocladospolide F (**56**). Among them, **54** and **55** constructed an unusual dimeric structure: a 12-membered lactone monomer and a hydrolyzed monomer condensed via an L-cysteine unit. Macrolides are a kind of natural antibiotic with significant antimicrobial activity, and some of them have been used as clinical drugs, for example, roxithromycin. In the bioassays, **52** exhibited potential antibacterial activity against the aquatic pathogen *Edwardsiella tarda* (MIC = 4 μg/mL), while compounds **48**–**51** displayed significant activities against the tested microorganisms, including the plant-pathogenic fungi *Colletotrichum glecosporioides*, *Fusarium oxysporum*, etc. [43,44].

**Figure 9 marinedrugs-20-00765-f009:**
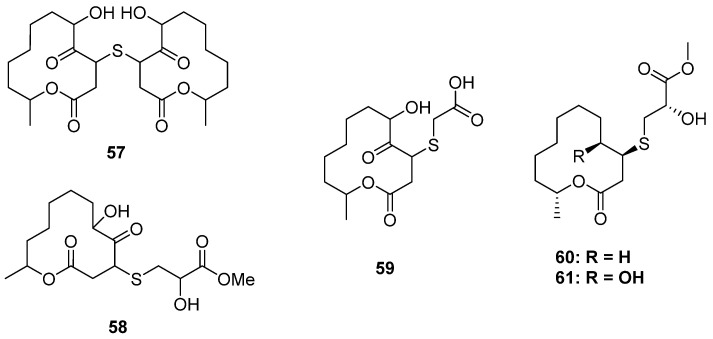
Chemical structures of **57**–**61**.

#### 2.2.3. 3-Thioether-Substituted Macrolides

Two-thirds of the S-containing 12-membered macrolides were substituted by a S atom at C-2, while the others were substituted at C-3. Pandangolides 2–4 (**57**–**59**) were C-3 thio-substituted 12-membered macrolides isolated from two sponge-derived fungi (Figure 9) (one was unidentified [45], and the other was *Cladosporium herbarum* [46]). However, their stereochemistry was still unknown, and no bioactivity test was carried out. The strain of *Cladosporium cladosporioides* MA-299 from *Bruguiera gymnorrhiza* not only yielded four C-2 thio-substituted macrolides, thiocladospolides A–D, but also produced two C-3 thio-substituted analogs, thiocladospolides F and G (**60** and **61**) [47]. In this study, biosynthesis was proposed, which suggested that the different position of sulfur substitution was due to the different way of nucleophilic addition between the cysteine and the lactone core. Compound **60** exhibited antifungal effects against the plant-pathogenic fungus *Helminthosporium maydis* with an MIC value of 4.0 μg/mL.

### 2.3. Cytochalasin

Cytochalasans are a group of fungal metabolites derived by polyketidenonribosomal peptide synthetase (PKS-NRPS). Cytochalasans usually feature a perhydro-isoindolone core fused with a macrocyclic ring and exhibit high structural diversities. Moreover, they have a broad spectrum of bioactivities, such as antimicrobial and cytotoxic activities. To date, though more than 500 cytochalasans have been discovered from different fungi, the S-containing analogs are quite rare (Figure 10 and Figure 11) [48].

Two epimeric cytochalasin dimers via thioether bridge thiocytochalasins C and D (**62** and **63**) were isolated from the endophytic fungus *Phoma multirostrata* [48]. Both of them exhibited strong cytotoxicity against five tested human cancer cell lines, MCF-7, HepG2, CT26, HT-29, and A549, with IC_50_ values from 0.76 to 7.52 μM. Moreover, they can significantly arrest the cell cycle G2/M phase of CT26 cells at a concentration of 1 μM. Another deep-sea squat lobster *Shinkaia crosnieri*-derived fungus, *Curvularia verruclosa,* also produced thiocytochalasin C, which was reported as a new metabolite verruculoid A at the same time. The bioassays of verruculoid A suggested that it displayed antibacterial activity against the human pathogenic *Escherichia coli* with an MIC of 2 μg/mL [49]. Except for the two dimers, the fungus *P. multirostrata* also yielded two monomers, thiocytochalasins A and B (**64** and **65**), featuring a novel 5/6/14/5 tetracyclic framework, which was driven from the cytotochalasin core and thioglycolic acid unit. However, **64** and **65** only exhibited weak cytotoxicity against HepG2 cells compared to the dimers.

**Figure 11 marinedrugs-20-00765-f011:**
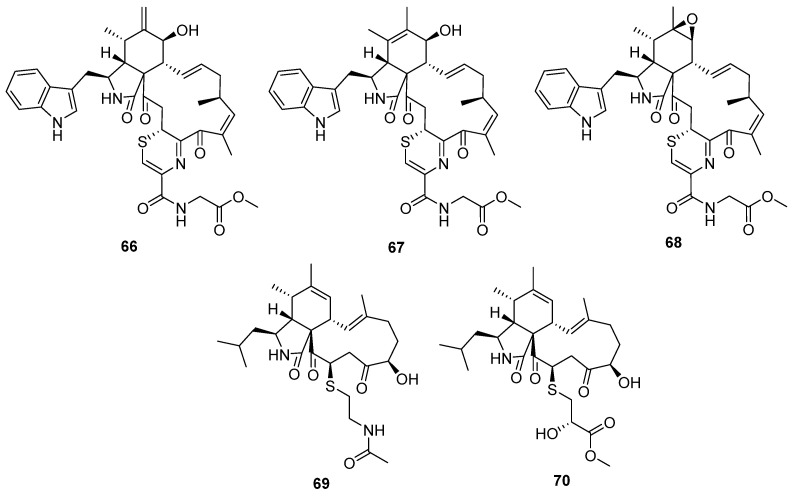
Chemical structures of **66–70**.

Fungal co-culture has been a new effective strategy to dig out structurally unique and bioactive metabolites from fungi. The co-culture of two terrestrial fungi, *Chaetomium globosum* and *Aspergillus flavipes,* led to the isolation of three unprecedented merocytochalasans, cytochathiazines A–C (**66** to **68**) [50], which represent the first examples of natural products featuring a 2H-1,4-thiazine moiety (Figure 11). The proposed biosynthesis pathway suggested that the special 2H-1,4-thiazine moiety was driven by a Michael addition between the cytochalasin core and dipeptide (cysteine/glycine). In the bioassays, compound **63** exhibited moderate antiproliferative activities against NB4 and HL-60 cell lines and induced moderate apoptosis by activation of caspase-3 and degradation of PARP as well. Cyschalasins A and B (**69** and **70**) were two cysteine-combined merocytochalasans from the endophytic fungus *Aspergillus micronesiensis* with moderate cytotoxic and antimicrobial activities (Figure 11) [51].

### 2.4. Alkaloids

The clinic antibiotics penicillin and cephalosporin C (Figure 12) are self-evident S-containing β-lactam derivatives isolated from *Penicillium* sp. and *Acremonium chrysogenum*, respectively [52,53]. Moreover, 20 S-containing alkaloids have been isolated from marine or terrestrial fungi (Figure 13, Figure 14 and Figure 15), excluding cytochalasin, which has been discussed above. The structures of S-containing alkaloids exhibited high structural diversity, such as amide, imide, pyridine, etc. Except for thioether, several alkaloids introduce the S atom in a special way, for example, the disulfide bond.

Robert and his co-workers collected a fungal strain of *Aspergillus unilateralis* MST-F8675 from a soil sample near Mount Isa, Queensland, and the chemical investigation of this fungus led to the isolation of three highly modified dipeptides, aspergillazines A–C (**71** to **73**), driven by two molecular phenylalanines [54]. All of them constructed a unique sulfur-bridged heterocyclic system, and **72**/**73** are the C-2 epimers derived from compound **71** via reductive ring opening of the 1,2-oxazine (Figure 13). A novel lumazine peptide, penilumamide (**74**), constructing an unusual 1,3-dimethyl-lumazine-6-carboxylic acid unit, a methionine sulfoxide unit, and an anthranilic acid unit, was isolated from marine-derived fungus *Penicillium* sp. [55]. Another marine-derived *Aspergillus* sp. (collected from gorgonian) also yielded two analogs, penilumamides B and C (**75** and **76**) (Figure 13) [56]. It is worth noting that the yields of compound **75** increased in the feeding culture with L-methionine of this strain, but it was unstable and easy to be oxidized into **74** or **76** when exposed to air. However, in the bioassays, the six alkaloids discussed above exhibited no bioactivities.

The chemical investigation of a marine sponge-derived fungus by Li et al. led to the identification of a nitrogen-containing thiophenone derivative, **77** (Figure 14). This was the first example of natural products containing a thiolactone moiety [57]. Lin and his co-workers discovered five new methylsuccinimide derivatives, violaceimides A–E (**78** to **82**), from the sponge-associated fungus *Aspergillus violaceus*, which were found in nature for the first time (Figure 14) [58]. All of them constructed a methylsuccinimide incorporating one or two modified cysteine units via a S atom, of which compound **78** was a dimer via a disulfide linkage. The stereochemistry of C-7 was established by Snatzke’s method and Mosher’s esterification, while the absolute configuration of methyl in the succinimide ring was deduced through acidic hydrolysis. Compounds **78** and **79** exhibited potential inhibition against acute monocytic leukemia U937 and human colonic HCT-8 with IC_50_ values ranging from 1.5 to 5.3 μM without any toxic effect on normal cells. Moreover, compound **82** also showed moderate activities against U937 cells with suitable selectivity. The preliminary structure–activity relationship analysis indicated that the 2-hydroxy3-mercaptopropanic unit plays an important role in cytotoxicity activity, while the introduction of a S atom might contribute to high selectivity. Three pyridine *N*-oxide dimers with disulfide linkage (**83** to **85**) were isolated from the basidiomycete *Cortinarius* sp. (Figure 14) [59]. Compounds **83** and **85,** containing 2-thiopyridine *N*-oxide functionality, exhibited significant cytotoxicity and antimicrobial activity (the details of the bioactivity results were not given in the published paper).

**Figure 14 marinedrugs-20-00765-f014:**
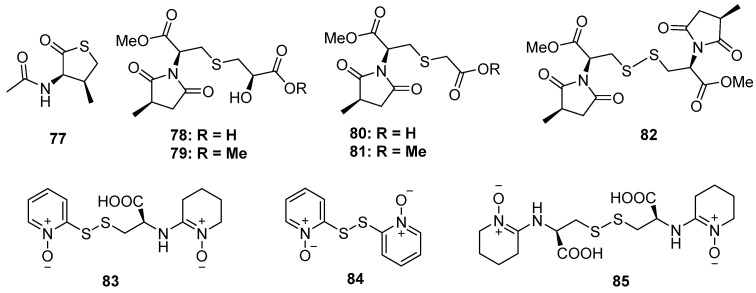
Chemical structures of **77–85**.

**Figure 15 marinedrugs-20-00765-f015:**
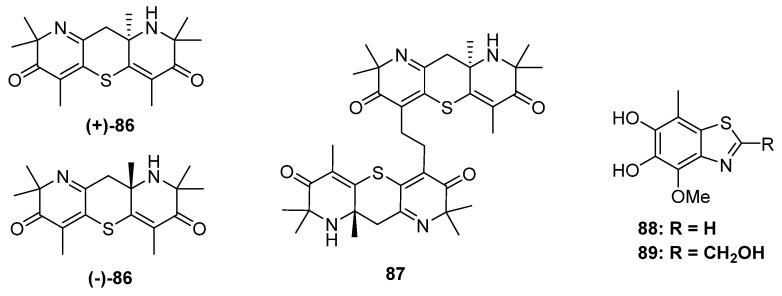
Chemical structures of **86**–**89**.

The research group of Liu has focused on the chemical and biological diversity of terrestrial fungi, especially the higher fungi. *Xylaria longipe* was a wood-decay fungus belonging to ascomycota, which was collected by Liu and his co-workers from southwest China. The chemical investigation of this strain led to the isolation of a piperidine derivative together with its dimer xylaridines C and D (**86** and **87**) (Figure 15). The monomer **86** possesses two piperidine units fused through a thiopyran ring with a chiral center at C-8. Thus, two optical pure enantiomers, (+)- and (−)-**86,** were separated and identified by X-ray diffraction. As for **87**, there should be four configurations since the monomer was enantiomeric. However, only (+)-**87** and a mixture of (−)-**87** and meso-**87** were obtained. (−)-and (+)-**86** exhibited moderate cytotoxicities against the MCF-7 cell line with IC_50_ values of 20.0 μM and 22.5 μM, respectively [60]. The strain *A. fumigatus* GZWMJZ-152 not only produced three hydrolyzed xanthones but also yielded two benzothiazoles with DPPH radical-scavenging activity (**88** and **89**) (Figure 15) [24].

## 3. Diversity Analysis

Overall, except for the two S-containing chromones from unknown source-derived fungi, more than half of the new S-containing NPs (excluding the TDPs) were isolated from marine fungi (45 were isolated from marine-derived fungi and 42 from terrestrial fungi) (Figure 16). The conclusion was also tenable if the TDPs were taken into consideration. Moreover, it could be concluded that the S-introduced macrolides are only discovered in marine fungi up to now, which occupied half of the marine fungi-derived S-containing metabolites. Then, the second largest class is polyketides, which contributed 27% of the total number of marine fungi and 52% of terrestrial fungi. The above results are consistent with the conclusion of the review of natural products from microorganisms, which points out that polyketides are still the largest class of fungal metabolites. Moreover, the S-containing cytochalasins are abundant in terrestrial fungi.

A chronological analysis (Figure 17) indicated that only a few S-containing metabolites (excluding the TDPs) were discovered in fungi before 2000. Since 2010, the number, especially that of the S-containing compounds from marine-derived fungi, has had dramatic growth. After 2020, the number also exhibited a rapid growth trend. This is probably because marine fungi have become a research hotspot in the past two decades, and advanced research methods such as metabolomics and genomics largely improved the isolation efficiency.

The species of fungi that produce S-containing metabolites are diverse (Figure 18). A total of 16 genera are included, belonging to *Aspergillus*, *Cladosporium*, *Penicillium*, *Talaromyces*, *Curvularia, Coniochaeta, Preussia, Hohenhuehelia, Alternaria, Panax, Phomopsis*, *Xylaria, Cortinarims, Neosartorya, Pseudallescheria,* and *Eruotium*. Among them, *Aspergillus* and *Penicillium* are still the most contributed species, which produce nearly half of the metabolites (40 compounds, accounting for 45%). It is noticed that *Cladosporium* sp. is an important producer only collected from a marine environment, which metabolizes 14 S-containing macrolides, accounting for 15% of the total compounds. Except for the *Asgergillus, Penicillium, and Cladosporium* species, the remaining 13 genera contribute a total of 36 compounds. Among them, *Talaromyces* spp. and *Phomopsis* spp. are the major producers (six compounds and five compounds, respectively).

Bioassays of the new S-containing NPs revealed that there are 60 metabolites (40% of the total) exhibiting various activities, including cytotoxic, antimicrobial, antioxidative, anti-inflammatory, *a*-glucosidase inhibitory, and lipid-lowing effects (Figure 19). Among them, the antimicrobial and cytotoxic activities are the most significant pharmacological activity, with 48 compounds exhibiting in vitro cytotoxicity against different tumor cell lines, such as A549, HT1080, U937, etc., or antimicrobial activity against pathogenic bacteria/fungi. S-containing macrolides could be a useful source of promising antibiotics compared to the other isolated metabolites since 16 of the 24 isolated macrolides displayed potential antimicrobial or cytotoxic activities. In addition, S-containing cytochalasins might be potential lead compounds, as 7 of 9 isolated metabolites showed cytotoxicity (three of which also showed antimicrobial effects). The other polyketides exhibited bioactive diversity, with three-quarters of the compounds exhibiting one or more types of bioactivities mentioned above.

## 4. Conclusions and Outlook

This review presents an overview of 89 new fungal-derived S-containing metabolites from 1929 to 2022, which mainly focuses on their fungal origin, chemical structures, and bioactivities. Though S-containing NPs producing fungi are distributed around the whole environment, marine fungi are important contributors of S-containing NPs since they contribute more than half of the total number. Moreover, due to the extreme survival environment and complicated metabolic mechanisms, marine fungi could produce different metabolites with significant bioactivities compared to terrestrial fungi, for example, antimicrobial macrolides. Therefore, marine fungi possess a huge potential to discover new bioactive S-containing metabolites and develop S-containing lead compounds.

The biosynthesis mechanisms of the C-S bond formation and how the sulfur element was introduced into secondary metabolites are the key and difficult points of S-containing NPs research. In the primary metabolites, persulfidic sulfur and the thiocarboxylate group on sulfur-donor proteins are essential sulfur sources, while the sulfur-introducing mechanisms in secondary metabolites might be diverse but remain unclear. Peptides driven from S-containing amino acids can easily construct thioesters under catalyzation by hydrolase. S-methyltransferases and glutathione-S-transferases catalyzing the attack of a thiol to activated carbon is one of the important ways to form the C-S bond in natural products such as lincomycin A [61,62], collismycin [63,64], and epipolythiodiketopiperazines. Moreover, some oxygenases, such as cytochrome P450 monooxygenases and flavoenzymes, can directly catalyze the connection between the S-containing amino acids and biosynthetic intermediates via S atom [65]. Some polypeptides can form the C-S bonds via non-enzyme ways, such as subtilosin A [66] and cyclothiazomycin [67]. A systematic review published by Hertweck et al. provided a comprehensive summary of C-S bond formation in natural products [68]. It could be concluded from this review that C-S formation in the biosynthesis of S-containing metabolites from bacteria or plants is studied in depth, while research on fungi is quite rare. Except for thiodiketopiperazines, which were driven from glutathione (Figure 20), there is no systematic study about the biosynthesis of S-containing NPs in fungi. Even though some metabolites listed in this review can be easily speculated to introduce the S atom by combining with cysteine, such as **1**, **2**, **13,** and most of the macrolides, etc., the other metabolites are still unknown. Therefore, it is necessary to carry out a systematic investigation of the biosynthesis of S-containing metabolites and dig out the different mechanisms of C-S bond formation, which will provide a new strategy to develop organosulfur drugs or lead compounds from the fungal resource.

In the bioassays, although nearly half of the S-containing metabolites from fungi exhibit potential pharmacological properties, a few of them were selected for further development in clinical application. Therefore, not only the biosynthesis pathway but also the deeper biological mechanisms of fungal-derived organosulfur NPs should be put on the agenda in the future.

## Figures and Tables

**Figure 1 marinedrugs-20-00765-f001:**
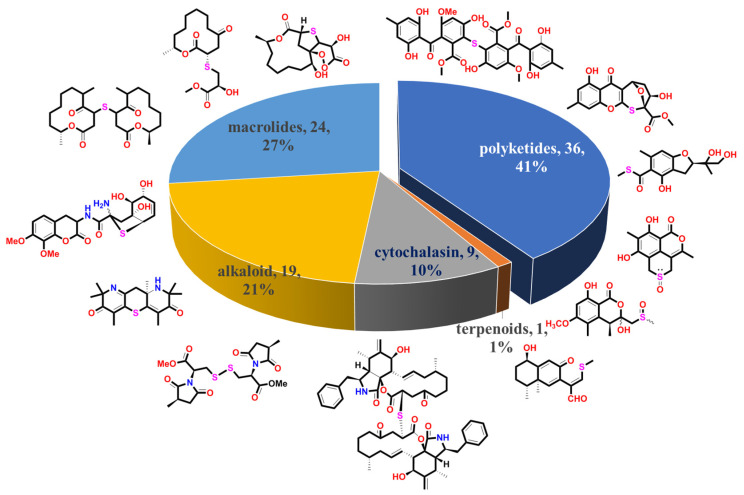
A total of 89 S-containing NPs were divided into five main groups.

**Figure 10 marinedrugs-20-00765-f010:**
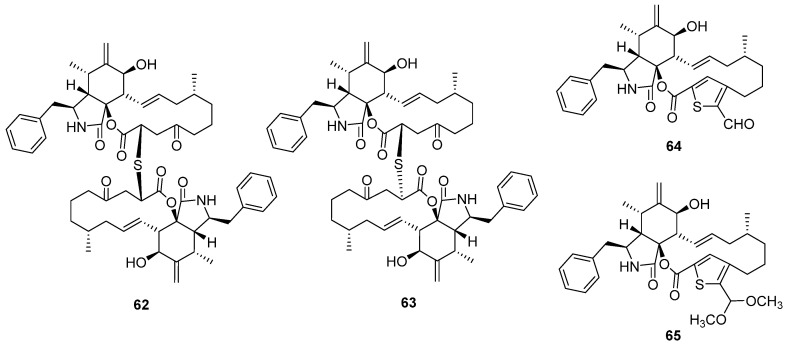
Chemical structures of **62–65**.

**Figure 12 marinedrugs-20-00765-f012:**
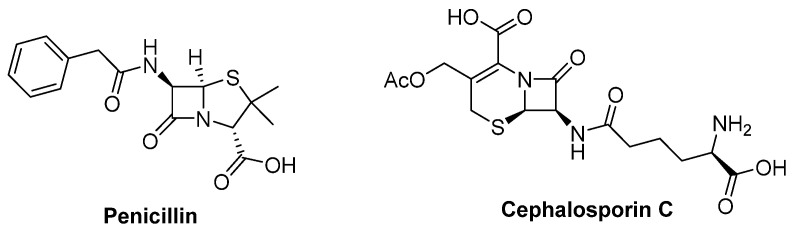
Chemical structures of penicillin and cephalosporin C.

**Figure 13 marinedrugs-20-00765-f013:**
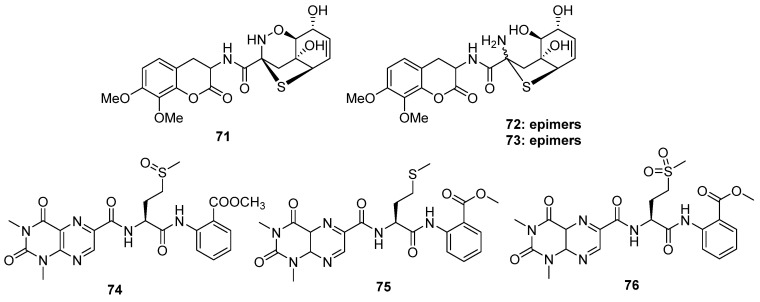
Chemical structures of **71–76**.

**Figure 16 marinedrugs-20-00765-f016:**
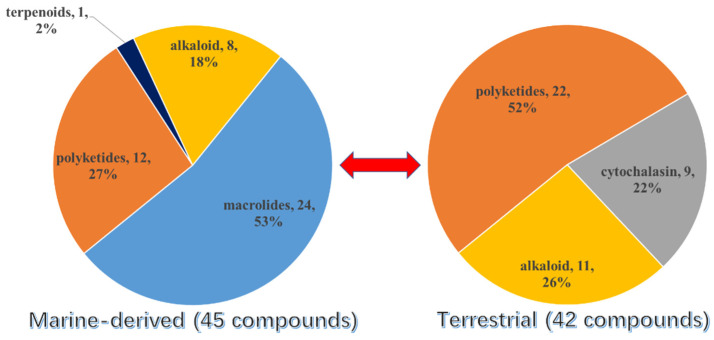
The source of S-containing metabolites with different structural classes.

**Figure 17 marinedrugs-20-00765-f017:**
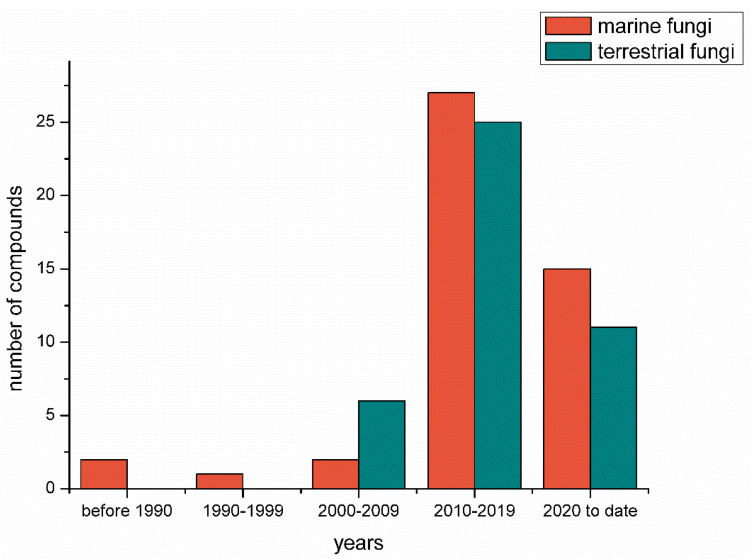
The number of S-containing metabolites (excluding TDPs) isolated from marine-derived and terrestrial fungi counted chronologically.

**Figure 18 marinedrugs-20-00765-f018:**
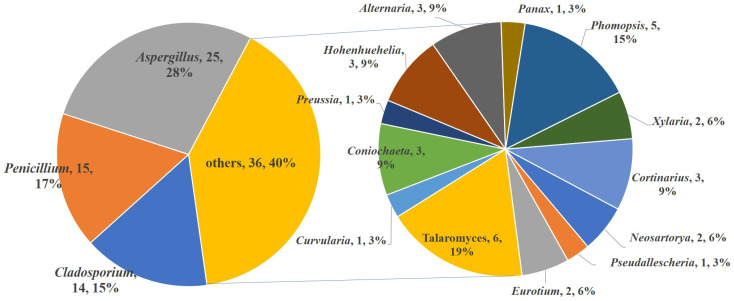
The summary of the fungal species producing S-containing metabolites.

**Figure 19 marinedrugs-20-00765-f019:**
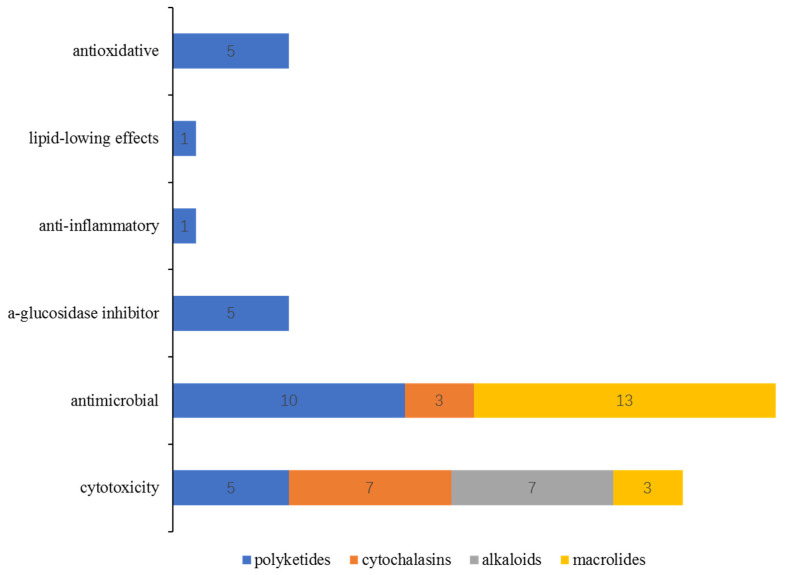
The relationship between the chemical features of S-containing NPs and their bioactivities.

**Figure 20 marinedrugs-20-00765-f020:**
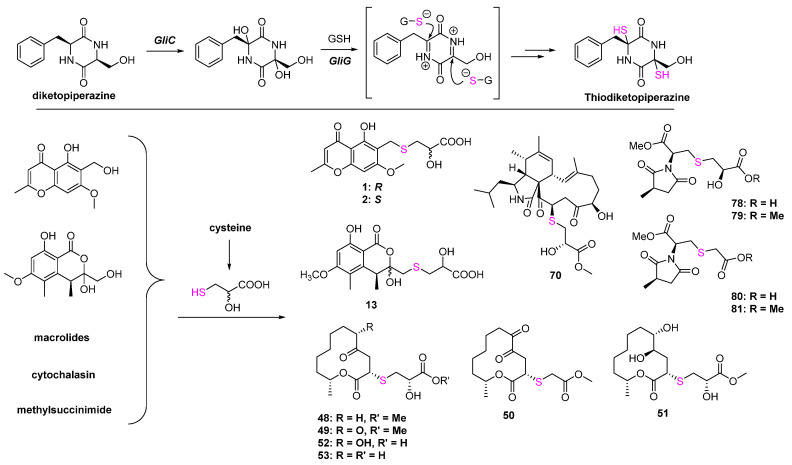
The known biosynthesis of thiodiketopiperazine and the speculated biosynthesis pathways of S-containing metabolites driven from cysteine.

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
