# Peer review of "Sulfur-Containing Metabolites from Marine and Terrestrial Fungal Sources: Origin, Structures, and Bioactivities"

_marinedrugs, 2022, doi:10.3390/md20120765_

Round 1
Reviewer 1 Report
The manuscript entitled “Sulfur-containing Metabolites from Marine and Terrestrial Fungal Sources: Origin, Structures and Bioactivities” by Liu et al. summarized 86 sulfur-containing secondary metabolites from fungi, and their biosynthetic pathways and biological activities were briefly described and evaluated. The structural and bioactive diversity of these S-containing metabolites were concluded in detail and the preliminary mechanism for C-S bond formation in fungi was also discussed briefly. The writing of the paper is relatively standard and well organized. However, the references of this review are incomplete, eg. the reference entitled “Sulfur-Containing Cytotoxic Curvularin Macrolides from Penicillium sumatrense MA-92, a Fungus Obtained from the Rhizosphere of the Mangrove Lumnitzera racemose” published in Journal of Natural products, (dx.doi.org/10.1021/np400614f | J. Nat. Prod. 2013, 76, 2145−2149) which reported three new sulfur-containing curvularin derivatives, was not cited in this review. In addition, there are some grammatical and spelling errors in the text, which need to be carefully checked and corrected by the authors.
Some details and grammatical errors are listed as below:
1 In line 84 of page 3, “The Chaetomium are a potential fungal genus which can produce diverse bioactive metabolites” should be “Chaetomium is a potential fungal genus that can produce diverse bioactive 84 metabolites”.
2 In line 117 of page 4, “yield” should be “yielded”.
3 In line 129 of page 4, “a ORAC” should be "an ORAC".
4 In line 141 of page 6, please delete “the” between in and sulfur-containing.
5 In line 157 of page 6, please add “a” between though and series.
6 In line 161 of page 6, “coral associated” should be “coral-associated”.
7 In line 165 of page 6, “constructing” should be “constructed”, “the” should be “a”.
8 In line 175 of page 6, “exmaple” should be “examples”.
9 In line 258 of page 9, please add “the” between arrest and cell.
10 In line 269 of page 9, “coculture” should be “co-culture”.
11 In line 287 of page 10, “have” should be “has”.
12 In line 289 of page 10, please delete "the" between as and amide.
13 In line 319 of page 11, “compounds” should be “compound”.
14 In line 322 of page 11, “play” should be “plays”.
15 In line 335 of page 12, “unit” should be “units”.
16 In line 389 of page 14, please add “are” between fungi and distributed.
17 In line 393 of page 14, “significantly” should be “significant”.
18 In line 400 of page 14, “difficul” should be “difficult”, “point” should be “points”.
19 In line 411 of page 14, “way” should be “ways”.
20 In line 415 of page 14, “are” should be “is”.
21 In line 416 of page 14, “was” should be “were”.
Author Response
Recommendation: revision.
The manuscript entitled “Sulfur-containing Metabolites from Marine and Terrestrial Fungal Sources: Origin, Structures and Bioactivities” by Liu et al. summarized 86 sulfur-containing secondary metabolites from fungi, and their biosynthetic pathways and biological activities were briefly described and evaluated. The structural and bioactive diversity of these S-containing metabolites were concluded in detail and the preliminary mechanism for C-S bond formation in fungi was also discussed briefly. The writing of the paper is relatively standard and well organized. However, the references of this review are incomplete, eg. the reference entitled “Sulfur-Containing Cytotoxic Curvularin Macrolides from Penicillium sumatrense MA-92, a Fungus Obtained from the Rhizosphere of the Mangrove Lumnitzera racemose” published in Journal of Natural products, (dx.doi.org/10.1021/np400614f | J. Nat. Prod. 2013, 76, 2145−2149) which reported three new sulfur-containing curvularin derivatives, was not cited in this review. In addition, there are some grammatical and spelling errors in the text, which need to be carefully checked and corrected by the authors.
Responds: Thank you very much for your useful advice. It was our carelessness to miss the reference published by Meng et al. in J Nat Prod and we have added the above reference in the revised manuscript and updated the data in the Diversity analysis part. Thank you again for your suggestion.
Some details and grammatical errors are listed as below:
1) In line 84 of page 3, “The Chaetomium are a potential fungal genus which can produce diverse bioactive metabolites” should be “Chaetomium is a potential fungal genus that can produce diverse bioactive metabolites”.
Responds: Thank you very much for your useful advice. We have revised the grammatical error according to your suggestion.
2) In line 117 of page 4, “yield” should be “yielded”.
Responds: Thank you for your helpful suggestion. We have revised the spelling error according to your suggestion.
3) In line 129 of page 4, “a ORAC” should be "an ORAC".
Responds: Thank you for your helpful suggestion. We have deleted the "a" here.
4) In line 141 of page 6, please delete “the” between in and sulfur-containing.
Responds: Thank you for your kind suggestion. We have deleted "the" here according to your suggestion.
5) In line 157 of page 6, please add “a” between though and series.
Responds: Thank you for your helpful suggestion. We have revised the grammatical error according to your suggestion.
6) In line 161 of page 6, “coral associated” should be “coral-associated”.
Responds: Thank you for your kind suggestion. We have revised the spelling mistake according to your suggestion.
7) In line 165 of page 6, “constructing” should be “constructed”, “the” should be “a”.
Responds: Thank you for your kind suggestion. We have revised the spelling mistake according to your suggestion.
8) In line 175 of page 6, “exmaple” should be “examples”.
Responds: Thank you for your helpful suggestion. We have revised the spelling mistake according to your suggestion.
9) In line 258 of page 9, please add “the” between arrest and cell.
Responds: Thank you for your helpful suggestion. We have added “the” between arrest and cell.
10) In line 269 of page 9, “coculture” should be “co-culture”.
Responds: Thank you for your helpful suggestion. We have changed “coculture” to “co-culture.
11) In line 287 of page 10, “have” should be “has”.
Responds: Thank you for your helpful suggestion. We have revised the grammatical error according to your suggestion.
12) In line 289 of page 10, please delete "the" between as and amide.
Responds: Thank you for your helpful suggestion. We have deleted "the" here according to your suggestion.
13) In line 319 of page 11, “compounds” should be “compound”.
Responds: Thank you for your suggestion. We have changed “compounds” to “compound”.
14) In line 322 of page 11, “play” should be “plays”.
Responds: Thank you for your suggestion. We have changed “play” to “plays”.
15) In line 335 of page 12, “unit” should be “units”.
Responds: Thank you for your suggestion. We have changed “unit” to “units”.
16) In line 389 of page 14, please add “are” between fungi and distributed.
Responds: Thank you for your suggestion. We have added "are" here according to your suggestion.
17) In line 393 of page 14, “significantly” should be “significant”.
Responds: Thank you for your suggestion. We have changed “significantly” to “significant”.
18) In line 400 of page 14, “difficul” should be “difficult”, “point” should be “points”.
Responds: Thank you for your suggestion. We have revised the grammatical and spelling errors according to your suggestion.
19) In line 411 of page 14, “way” should be “ways”.
Responds: Thank you for your suggestion. We have changed “way” to “ways”.
20) In line 415 of page 14, “are” should be “is”.
Responds: Thank you for your suggestion. We have revised the grammatical error according to your suggestion.
21) In line 416 of page 14, “was” should be “were”.
Responds: Thank you for your suggestion. We have revised the grammatical error according to your suggestion.
Reviewer 2 Report
First of all, this is a weak review with regards of its content. 50% of the content of the review are coming from terrestrial organisms, which is not related to the journal and not of interest to the readers of this journal.
I suggest the review should be adjusted more to adapt the "Marine Drugs" after removal of the non-relevant contents and enriching the review with related compounds from the literature.
Moreover, the author did not mention in the review how they did the search to find all S-containing compounds or which search engine ( for example SciFinder), or how they filter the collected data or literature and how they dereplicate the results of the search.
In conclusion, the review is a weak one and should be completely revised to address the above points and meets the requirements of the journal and attract the readers of Marine Drugs
Author Response
First of all, this is a weak review with regards of its content. 50% of the content of the review are coming from terrestrial organisms, which is not related to the journal and not of interest to the readers of this journal.
I suggest the review should be adjusted more to adapt the "Marine Drugs" after removal of the non-relevant contents and enriching the review with related compounds from the literature.
Moreover, the author did not mention in the review how they did the search to find all S-containing compounds or which search engine (for example SciFinder), or how they filter the collected data or literature and how they dereplicate the results of the search.
In conclusion, the review is a weak one and should be completely revised to address the above points.
Responds: Thank you for your helpful advice. Though the scope of Marine Drugs is to publish the research results of the discovery, development, exploitation, and production of biologically and therapeutically active compounds from marine habitats, We think it is valuable to make the comparison between marine and terrestrial S-containing natural products, which was more readable and straightforward for the readers to know the special of marine S-containing natural products compared to those from terrestrial fungi. Thus, we did not remove the terrestrial part.
As to the literature searching, we used the Scifinder and Google Scholar website to search the S-containing compounds using different keywords such as "Sulfur-containing" and "thio". Moreover, we have also double-checked the different databases which contain journals publishing novel natural products such as the Elsevier and ACS.
Reviewer 3 Report
This review compiles the chemical structures and biological activities of 42 marine fungal-derived S-containing metabolites and 42 terrestrial fungal-derived S-containing metabolites, isolated between isolated from 1929 to the present.
The scope of the review could be better clarified in the abstract and introduction. Reading lines 42-43 it seems that this review includes S-containing metabolites that were isolated in marine or terrestrial from 1929 to the present, but in fact, regarding S-containing NPs from marine derived fungi, this review is not comprehensive as only reports the marine fungal-derived S-containing metabolites that were not included in a recent and systematic review [ref10: Mar. Life Sci. Technol. 2021, 3, 488-518] which compiled 213 S-containing NPs from marine derived fungi from 1987 to 2020. Regarding terrestrial fungal-derived S-containing metabolites it may be more clear the scope of the study. Are the 42 terrestrial fungal-derived S-containing metabolites, all the terrestrial fungal-derived S-containing metabolites isolated between 1929 to the present?
In the final critical analysis, aimed to conclude about S-containing NPs producing fungi from 1929 to the present, the authors conclude: “Though the S-containing NPs producing fungi distributed around the whole environment, the marine fungi are the important contributors of S-containing NPs since they contribute half of the total number.” The 213 S-containing NPs from marine derived fungi, compiled in ref10, cannot be forgotten in this conclusion. Taken in account, the total contribution of marine fungi is much more than half of the terrestrial. Also regarding the terrestrial fungal-derived S-containing metabolites, it is not clear if there were more than the 42 terrestrial fungal-derived S-containing metabolites described in this work, and they must be taken in consideration in the final discussion.
Also regarding the final analysis it would be much appreciated to see a chronological analysis by decades (for examples graphical bars 1929-1939/ 1939-1949/ etc) of the S-containing fungi metabolites, to support the phrase in the abstract that states: “In recent years, an increasing number of S-containing metabolites have been discovered (…)”. Having in mind the readers of Marine Drugs, it would also be interested to see a chronological analysis of the discovered S-containing marine fungi metabolites in contrast to the S-containing terrestrial fungi metabolites.
Minor:
Line 11: Add the abbreviation (S) following the first time the Sulfur word appears and replace “natural products” by NPs (the abbreviation NPs was previous defined in line 10).
Line 33: Replace “In resent years” with “In recent years”.
Line 42: Replace “focuing” with “focusing”.
Line 51-52 and Line 75: Replace “sulfur-containing natural products” with “S-containing NPs”.
Line 68: Replace “sulfur atom” with “S atom”.
Lines 148-152: the docking study is missing the target used.
Line 192: add under brackets the number of the cited “thioglycolic acid thioester derivative”.
Figures must be cited along the text within sections 2.1.1-3 and 2.2.1-3.
R is missing in the second structure of Figure 4.
In figure 5, change the place of compound 25 and 26 to the place where compounds 27 and 28 are.
In figure 6, change the place of compounds 34R and 34S to the place where compound 35 is.
In figure 8, again, the compounds are not placed in numerical order: change the compound 53 to the end.
In section 2.2.3, intitled “3-thioether-substituted macrolides”, compounds 56 and 57 are being mentioned as C-2-thiosubstituted analogues (lines 239-240). Please revise.
Figure 12 and 16 are placed before being cited in the text. Please move Figure 12 and 16 in order to be shown after the citation.
Some references are missing the titles. See ref 29-35.
Author Response
This review compiles the chemical structures and biological activities of 42 marine fungal-derived S-containing metabolites and 42 terrestrial fungal-derived S-containing metabolites, isolated between isolated from 1929 to the present.
The scope of the review could be better clarified in the abstract and introduction. Reading lines 42-43 it seems that this review includes S-containing metabolites that were isolated in marine or terrestrial from 1929 to the present, but in fact, regarding S-containing NPs from marine derived fungi, this review is not comprehensive as only reports the marine fungal-derived S-containing metabolites that were not included in a recent and systematic review [ref10: Mar. Life Sci. Technol. 2021, 3, 488-518] which compiled 213 S-containing NPs from marine derived fungi from 1987 to 2020. Regarding terrestrial fungal-derived S-containing metabolites it may be more clear the scope of the study. Are the 42 terrestrial fungal-derived S-containing metabolites, all the terrestrial fungal-derived S-containing metabolites isolated between 1929 to the present?
In the final critical analysis, aimed to conclude about S-containing NPs producing fungi from 1929 to the present, the authors conclude: “Though the S-containing NPs producing fungi distributed around the whole environment, the marine fungi are the important contributors of S-containing NPs since they contribute half of the total number.” The 213 S-containing NPs from marine derived fungi, compiled in ref10, cannot be forgotten in this conclusion. Taken in account, the total contribution of marine fungi is much more than half of the terrestrial. Also regarding the terrestrial fungal-derived S-containing metabolites, it is not clear if there were more than the 42 terrestrial fungal-derived S-containing metabolites described in this work, and they must be taken in consideration in the final discussion.
Responds: Thank you very much for your useful advice. We have added the scope of this review in the abstract and introduction according to your suggestion.
The ref10 cited in this manuscript summarized 484 the sulfur-containing natural products from marine microorganisms from 1987 to 2020, which contain 213 compounds from marine-derived fungi. However, most of them are thiodioxopiperazines, and this review aimed to make it more straightforward to count the fungal-derived S-containing NPs and understand the particularity of marine S-containing natural products compared to those from terrestrial fungi. Since the TDPs are too much abundant in fungi and the the ref10 has given a comprehensive summary of this class of metabolites, we do not give a detailed discussion in this review. Furthermore, the polypeptides of which the building blocks include S-containing amino acids are also excluded. As a result, 45 marine fungal-derived S-containing NPs and 42 terrestrial fungal-derived ones were summarized. After the literature searching using the Scifinder and Google Scholar website as well as the different databases which contain journals publishing novel natural products such as the Elsevier and ACS. A total of 42 terrestrial fungal-derived S-containing metabolites are included, which exclude the TDPs and polypeptides (of which the building blocks include S-containing amino acids). Therefore, the data presented in the compound summary and the final critical analysis did not contain the TDPs and S-containing polypeptides. We have added some explanation in the Diversity analysis part. Thank you again for your suggestion.
Also regarding the final analysis it would be much appreciated to see a chronological analysis by decades (for examples graphical bars 1929-1939/ 1939-1949/ etc) of the S-containing fungi metabolites, to support the phrase in the abstract that states: “In recent years, an increasing number of S-containing metabolites have been discovered (…)”. Having in mind the readers of Marine Drugs, it would also be interested to see a chronological analysis of the discovered S-containing marine fungi metabolites in contrast to the S-containing terrestrial fungi metabolites.
Responds: Thank you very much for your useful advice. According to your suggestion, we counted the number of the S-containing metabolites from marine-derived and terrestrial fungi chronologically and added the corresponding data in the section of Diversity analysis. It could be concluded that only a few S-containing metabolites (excluded the TDPs) were discovered from fungi before 2000. Since 2010, the number, especially that of the S-containing compounds from marine-derived fungi, has a dramatical grow. After the year 2020, the number also exhibited a rapid growth trend. This is probably because the marine fungi have become a research hotspot in the past two decades and the advanced research methods such as the metabolomics and genomics largely improved the isolation efficiency.
Minor:
1) Line 11: Add the abbreviation (S) folloswing the first time the Sulfur word appears and replace “natural products” by NPs (the abbreviation NPs was previous defined in line 10).
Responds: Thank you very much for your useful advice. We have revised them according to your suggestion.
2) Line 33: Replace “In resent years” with “In recent years”.
Responds: Thank you for your helpful suggestion. We have revised the spelling error according to your suggestion.
3) Line 42: Replace “focuing” with “focusing”.
Responds: Thank you for your helpful suggestion. We have revised the spelling error according to your suggestion.
4) Line 51-52 and Line 75: Replace “sulfur-containing natural products” with “S-containing NPs”
Responds: Thank you for your kind suggestion. Thank you for your helpful suggestion. We have revised the mistake according to your suggestion.
5) Line 68: Replace “sulfur atom” with “S atom”.
Responds: Thank you for your helpful suggestion. We have used the abbreviation S according to your suggestion.
6) Lines 148-152: the docking study is missing the target used.
Responds: Thank you for your kind suggestion. The grammatical mistake has been revised according to your suggestion.
7) Line 192: add under brackets the number of the cited “thioglycolic acid thioester derivative”.
Responds: Thank you for your kind suggestion. The number of the compounds have been added according to your suggestion.
8) Figures must be cited along the text within sections 2.1.1-3 and 2.2.1-3.
Responds: Thank you for your helpful suggestion. The figures have been cited in the corresponding sections.
9) R is missing in the second structure of Figure 4.
Responds: Thank you for your helpful suggestion. The structure has been revised in Figure 4.
10) In figure 5, change the place of compound 25 and 26 to the place where compounds 27 and 28 are
Responds: Thank you for your helpful suggestion. The position of the structures in Figure 5 has been modified according to your suggestion.
11) In figure 6, change the place of compounds 34R and 34S to the place where compound 35 is
Responds: Thank you for your helpful suggestion. The position of the structures in Figure 6 have been modified according to your suggestion.
12) In figure 8, again, the compounds are not placed in numerical order: change the compound 53 to the end.
Responds: Thank you for your helpful suggestion. The position of the structures in Figure 8 have been modified according to your suggestion.
13) In section 2.2.3, intitled “3-thioether-substituted macrolides”, compounds 56 and 57 are being mentioned as C-2-thiosubstituted analogues (lines 239-240). Please revise.
Responds: Thank you for your suggestion. We have revised the spelling mistake in the description of compounds 56 and 57 (updated to compounds 60 and 61 in the revised manuscript)
14) Figure 12 and 16 are placed before being cited in the text. Please move Figure 12 and 16 in order to be shown after the citation.
Responds: Thank you for your suggestion. We have reformatted the manuscript and all the figures have been placed before being cited in the text.
15) Some references are missing the titles. See ref 29-35.
Responds: Thank you for your suggestion. The titles of ref 29 to 35 have been added.
Reviewer 4 Report
The topic of sulfur containing fungal metabolites is a very broad one. Nevertheless, the authors have written an excellent review that concisely and comprehensively describes the field, ranging from classic metabolites like penicillin to recent discoveries. The discussion includes comments regarding the structure elucidation, biosynthesis and bioactivity.
Overall, this is an important and timely review and I believe it will be well cited by the scientific community. I recommend the authors publish periodic updates in the future.
Author Response
The topic of sulfur containing fungal metabolites is a very broad one. Nevertheless, the authors have written an excellent review that concisely and comprehensively describes the field, ranging from classic metabolites like penicillin to recent discoveries. The discussion includes comments regarding the structure elucidation, biosynthesis and bioactivity. Overall, this is an important and timely review and I believe it will be well cited by the scientific community. I recommend the authors publish periodic updates in the future.
Responds: Thank you very much for your recognition of our work. We hope this review will make it more straightforward for readers to acquire the diversity of the fungal-derived S-containing NPs and understand the particularity of marine S-containing natural products compared to those from terrestrial fungi. We will also try our best to updated the data in the future.
Round 2
Reviewer 2 Report
The MS can be accepted in its current form
Author Response
Thank you very much for your recognition of our work. We hope this review will make it more straightforward for readers to acquire the diversity of the fungal-derived S-containing NPs and understand the particularity of marine S-containing natural products compared to those from terrestrial fungi. We will also try our best to updated the data in the future.
Reviewer 3 Report
I am happy to see that authors tried to address all the issues raised, which made this ms better, namely with the introduction of the chronological analysis by decades (Figure 17) where it was possible to put forward that the marine fungi have become a hotspot source in the past two decades of S-containing fungi metabolites in contrast to terrestrial ones.
The authors are advised to comment in the ms why sulfated metabolites were excluded and/or if the authors didn't find sulfated metabolites isolated from fungi.
Minor:
Abstract:
line 17-18: replace “to those from terrestrial fungi. We summarized the” with “to those from terrestrial fungi, we summarized the”
Introduction:
Line 48: replace “(…) to those from terrestrial fungi. We discussed” with “(…) to those from terrestrial fungi we discussed”
Figure 15:
Structure of compound 87 should be placed in the middle, between (+)86 and (-)86 and 88: switch strcutres (+)86 and (-)86 to the left of the figure.
Author Response
I am happy to see that authors tried to address all the issues raised, which made this ms better, namely with the introduction of the chronological analysis by decades (Figure 17) where it was possible to put forward that the marine fungi have become a hotspot source in the past two decades of S-containing fungi metabolites in contrast to terrestrial ones.
The authors are advised to comment in the ms why sulfated metabolites were excluded and/or if the authors didn't find sulfated metabolites isolated from fungi.
Responds: Thank you very much for your useful advice. Actually, the sulfated metabolites are a special kind of S-containing metabolites of which no C-S bond were constructed. The S atom in sulfated metabolites may probably introduced from the electrolyte of the culture (for example MgSO4) but not the primary metabolites. Thus, we did not conclude the sulfated metabolites in this MS. We have added the reason in the manuscript according to your suggestion.
Minor:
1) line 17-18: replace “to those from terrestrial fungi. We summarized the” with “to those from terrestrial fungi, we summarized the”
Responds: Thank you very much for your useful advice. We have revised spelling according to your suggestion.
2) Line 48: replace “(…) to those from terrestrial fungi. We discussed” with “(…) to those from terrestrial fungi we discussed”
Responds: Thank you for your helpful suggestion. We have revised the spelling error according to your suggestion.
3) Figure 15: Structure of compound 87 should be placed in the middle, between (+)86 and (-)86 and 88: switch structures (+)86 and (-)86 to the left of the figure.
Responds: Thank you for your helpful suggestion. We have revised the Figure 15 according to your suggestion.